# Psychomotor Performance after 30 h of Sleep Deprivation Combined with Exercise

**DOI:** 10.3390/brainsci13040570

**Published:** 2023-03-28

**Authors:** Tomasz Mikulski, Monika Górecka, Anna Bogdan, Magdalena Młynarczyk, Andrzej W. Ziemba

**Affiliations:** 1Mossakowski Medical Research Institute, Polish Academy of Sciences, 02-106 Warsaw, Poland; monikag@imdik.pan.pl (M.G.); ziemba@imdik.pan.pl (A.W.Z.); 2Faculty of Building Services, Hydro and Environmental Engineering, Warsaw University of Technology, 00-661 Warsaw, Poland; anna.bogdan@pw.edu.pl; 3Central Institute for Labour Protection—National Research Institute, 00-701 Warsaw, Poland; m.mlynarczyk@ciop.pl

**Keywords:** sleep deprivation, exercise, ultra-endurance, multiple-choice reaction time, psychomotor performance, central fatigue, adventure racing

## Abstract

Sleep deprivation (SD) usually impairs psychomotor performance, but most experiments are usually focused on sedentary conditions. The purpose of this study was to evaluate the influence of 30 h of complete SD combined with prolonged, moderate exercise (SDE) on human psychomotor performance. Eleven endurance-trained men accustomed to overnight exertion were tested twice: in well-slept and non-fatigued conditions (Control) and immediately after 30 h of SDE. They performed a multiple-choice reaction time test (MCRT) at rest and during each workload of the graded exercise test to volitional exhaustion. At rest, the MCRT was shorter after SDE than in the Control (300 ± 13 ms vs. 339 ± 11 ms, respectively, *p* < 0.05). During graded exercise, there were no significant differences in MCRT between groups, but the fastest reaction was observed at lower workloads after SDE (158 ± 7 W vs. 187 ± 11 W in Control, *p* < 0.05). The total number of missed reactions tended to be higher after SDE (8.4 ± 0.7 vs. 6.3 ± 0.8 in Control, *p* = 0.06). In conclusion, SDE is different from SD alone; however, well-trained men, accustomed to overnight exertion can maintain psychomotor abilities independently of the extent of central fatigue. Exercise can be used to enhance psychomotor performance in sleep-deprived subjects in whom special caution is required in order to avoid overload.

## 1. Introduction

Sleep deprivation (SD) entails a broad spectrum of negative physiological and psychological outcomes, with significant declines in psychomotor performance, cognition, vigilance and attention [1,2,3,4,5]. Few studies found no adverse effects of SD on psychomotor performance [6,7]. After 36 h of SD, the impaired reaction time was accompanied by redistribution in the cerebral blood flow, while no change in gray matter volume was found [4]. One night of SD adversely affected cognitive performance, which was correlated with changes in electroencephalography, especially in the occipital derivations [5].

There is sufficient evidence that physical activity has positive impact on cognitive functions in humans [8]. Even a single bout of aerobic exercise improves brain plasticity and positive effects are present for at least 30 min following exercise [9]. Enhanced cognition was observed during and after aerobic exercise lasting at least 20 min [10]. Training, especially interval of high intensity, was found to improve psychomotor performance [11]. Changes in cortical activation located in the left dorsal–lateral prefrontal cortex correlated with increased executive performance were also described [12]. The positive impact of exercise on reaction time is present regardless of age [13]. In studies on regeneration after spinal cord injury, exercise is documented to enhance brain plasticity by activation of neural pathways and neurotrophins: brain-derived neurotrophic factor (BDNF), neurotrophins 3 and 4, nerve growth factor and glial-cell-derived neurotrophic factor [14].

The relationship between psychomotor performance and exercise intensity resembles a U-shape: reaction time improves with increasing workloads and after exceeding the critical point (usually just beyond the anaerobic threshold) it breaks down rapidly [15,16,17,18].

The combined effect of SD and exercise on psychomotor performance has not been studied extensively yet [19]. Undoubtedly, they both affect it and the interrelation is complex; the final outcome of the measured task, e.g., reaction time, would rely on a subtle balance between stimulation and fatigue. It can be impaired or enhanced depending on conditions: type, intensity and duration of exercise, type of cognition test and besides sleep, all other factors influencing the arousal of subjects [20]. Therefore, for practical applications, only very closely simulated experiments (including both protocol and subjects) would be reproductive, whereas others will be just approximate and could generate misleading conclusions. Knowledge and awareness of the pattern of changes in psychomotor performance and underlying mechanisms, resulting from both sleep deprivation and exercise, would be useful in planning the strategy of task implementation in such activities (shift workers, ultra-endurance athletes) or enable its modification during operation.

The purpose of the present study was to evaluate the influence of 30 h of complete SD combined with prolonged exercise (SDE) on human psychomotor performance.

## 2. Materials and Methods

### 2.1. Participants

The study was conducted in 11 healthy male volunteers aged 31 ± 2 years with BMI 22.2 ± 0.3 kg/m^2^ (body mass 71 ± 1 kg; height 179 ± 2 cm) and peak oxygen uptake 56 ± 4 mL/kg/min. They were endurance-trained amateur athletes with an adventure racing experience (multi-discipline events involving overnight orienteering on foot, bicycles, kayaks). All subjects gave and signed their informed consent to participate in the study. The investigation was in accordance with the principles outlined in the Declaration of Helsinki and was approved by the Warsaw Medical University Ethics Committee, with the permission number KB/213/2010. 

### 2.2. Procedure

The sleep deprivation and exercise protocol (SDE) was a simulated adventure race that started very early in the morning at 5 a.m. (Figure 1). The whole study was performed in autumn in moderate and similar climate conditions (mean air temperature 9 °C). The subjects were equipped with proper clothing to allow for thermal comfort. They were trekking, mountain biking, running and orienteering in the forests and fields around Warsaw, kayaking on the Vistula river and inline skating on asphalt bicycle paths. Only a few short breaks between the different activities were allowed for a change of equipment and feeding. Exercise intensity was moderate (55–70% of the maximal individual heart rate) and continuously monitored with heart rate monitors (Forerunner 310 XT, Garmin, Olathe, KS, USA). The heart rate alert function was active and notified subjects if they exceeded the upper limit of the allowed intensity; in such case, they reduced their effort immediately. During the 30 h of exercise, subjects were not allowed to get any sleep. Laboratory tests were performed twice, both at approximately 11 a.m., in non-fatigued and well-slept conditions (Control) and immediately after SDE, in a random order with at least a one week interval between them. In the SDE trial, subjects ran or cycled to the laboratory and the onset of measurements was started after the change of clothes, rapid refeeding and installation of sensors, which all lasted approximately 30–45 min.

### 2.3. Psychomotor Performance

The multiple-choice reaction time test (MCRT) was measured at rest and during the graded exercise until volitional exhaustion, which was performed on a bicycle ergometer (EM 840, Siemens, Berlin, Germany) with the workloads increased by 50 W every 3 min starting with 50 W (Figure 1). The MCRT console producing the three color light signals and a sound was mounted on the wall 1.5 m in front of the cycloergometer at the eye level of the subjects (MRK 432, TEMED, Dąbrowa Górnicza, Poland). The reaction buttons were attached to the handlebars of the cycloergometer within the reach of the thumbs without the need to move hands on the grips. The test consisted of 15 positive (red light or sound) and 15 negative (green and yellow light) stimuli applied randomly in 1 to 4 s intervals. The subjects were asked to respond as quickly as possible by pressing the buttons with their thumbs: right hand in response to the red light and, the left hand in response to the sound and to not react to the negative stimuli. The total time for each MCRT trial was 107 s. The subjects were familiarized with the procedure a few days before starting the study by practicing the task both at rest and during exercise until no errors were made. The reaction time was determined at the nearest 0.01 s. The results are presented as the mean reaction time of 15 responses to positive stimuli. The missed reaction was defined as the lack of response or time longer than 700 ms. The shortest MCRT was calculated as the approx. 10% of the fastest positive responses; in our case, it was the average of the two fastest positive responses in each MCRT [4]. The MCRT test was conducted in a normal, well-slept condition (Control) and immediately after 30 h of SDE, at rest and during the last 2 min of each workload of the incremental, graded exercise test. 

### 2.4. Statistical Analysis

The data are presented as means with standard errors (SEM). The normality of the variables was analyzed using the Shapiro–Wilk test. Accordingly, the paired *t*-test was used for normally distributed results (MCRT, the shortest MCRT and the workload corresponding to the fastest MCRT), whereas the Wilcoxon test was used for non-normally distributed results (total number of missed reactions). Changes over time were examined with two-way analysis of variance (ANOVA) for repeated measures. The two factors were the SDE and the time of the repeated measurement of the MCRT or the shortest MCRT during each stage of the exercise tests. When a significant *F* value was obtained, a paired t-test was used to detect the pairwise differences between means. The workloads corresponding to the fastest MCRT were calculated individually from the MCRT graph for each subject during the graded exercise test with a polynominal fit curve. The level of significance was accepted at *p* < 0.05 and Statistica version 6 software (Statsoft Inc., Tulsa, OK, USA) was used for the calculations.

## 3. Results

At rest, MCRT was significantly shorter after SDE than in the Control (300 ± 13 ms vs. 339 ± 11 ms, respectively, *p* < 0.05; Figure 2). The shortest MCRT values were not different between the Control and SDE at rest (Figure 3). During graded exercise, there were no significant differences between trials in MCRT, as well as in the shortest MCRT. The fastest MCRT was observed at a significantly lower workload after SDE (158 ± 7 W vs. 187 ± 11 W in Control, *p* < 0.05). The total number of missed reactions tended to be higher after SDE (8.4 ± 0.7 vs. 6.3 ± 0.8 in Control, *p* = 0.06). ANOVA showed a significant effect of the time of the repeated measurement (workload) on MCRT in both the Control and SDE trials (*p* < 0.001). 

## 4. Discussion

The novel aspect of the present study reveals the psychomotor performance measured during exercise in subjects submitted to 30 h of sleep deprivation combined with moderate intensity prolonged exercise (SDE). Interestingly, psychomotor performance measured at rest in the SDE trial was improved, most probably because the additional stimulation induced by prolonged exercise annihilated the usually observed detrimental effect of sleep deprivation [2,3,4,5,19]. Several factors influenced by exercise also affect the brain and might have led to such a result. Participants in the present study were adapted to overnight exertion and tolerated 30 h of intermittent physical activity well; therefore, exhaustion of the hypothalamic–pituitary–adrenal axis is very unlikely. They rather experienced the general enhancement of arousal induced by the ultra-endurance exercise, producing stimulation of the hypothalamic–pituitary–adrenal axis with consecutive hormonal changes in catecholamines and ACTH, which are known to correlate with reaction time and the number of mistakes made [21]. Similar results of increased reaction speed due to arousal activation were found by Chang et al. and Audriffren et al. [22,23]. 

The possible background of the neurobiological mechanisms is presented in a review by Chen and Nakagawa [24]. The adaptations to exercise positively affecting brain plasticity may result from the enhanced exertional concentration of BDNF, insulin-like growth factor-1 and vascular-endothelial-derived growth factor, which stimulate angiogenesis and the growth and development of neurons. Lactate released by exercising skeletal muscles serves as an energy substrate for the brain cells, glutamate synthesis (the main excitatory neurotransmitter) and the maintenance of long-term potentiation [25,26,27]. The hippocampal neuronal network was found to mature more rapidly (as indicated by the synapse development and synchronous neuronal activity) if its cell culture was incubated in the media from contracting muscle cells [28]. The exertional release of anti-inflammatory cytokines, such as interleukin-6, interleukin-1 receptor antagonist and interleukin-10, reduces neuro-inflammation, whereas the enhanced activity of antioxidant enzymes and the number of mitochondria reduce oxidative stress, preventing the ROS-induced dysfunctions in the brain and brain–blood barrier [29,30]. Exercise stimulates the release of the neurotransmitters dopamine and serotonin, which influence neurogenesis and cognition. Treadmill running was reported to increase dopamine in the prefrontal cortex and improve spatial memory in sham-operation rats [31]. Serotonin plays a crucial role in the regulation of neurogenesis in the hippocampus of adults [32,33]. Pietrelli et al. have found that treadmill running increased serotonin and BDNF in the cortex and hippocampus, which was accompanied by better performance in an object recognition test [34]. All the aforementioned factors are also active during a single bout of physical activity and may contribute to psychomotor performance and arousal during exercise and recovery.

The other factor which increases while exercising at a moderate intensity is the cerebral blood flow [35,36]. Such findings were confirmed with non-invasive functional near-infrared spectroscopy, where even 10 min of exercise at 30% peak oxygen uptake were enough to induce significant improvements in cognitive performance and arousal levels [1]. Moreover, the difference in blood flow distribution between the internal and external carotid arteries should also be considered. If exercise is connected with a significant thermal load, the blood flow in the external carotid artery increases in order to prioritize thermoregulation and thus the flow in the internal carotid artery and the brain itself is decreased [37]. Diving deeper into the brain, interesting results were found regarding the relationship between blood flow in the middle cerebral artery and cognitive function, measured with a Stroop test during 50 min of exercise at the heart rate of 140 bpm [38]. Despite the decreased blood flow in the middle cerebral artery, the reaction times were improving and when blood flow was elevated by external hypercapnia, it did not affect psychomotor performance. Therefore, in those exercise conditions, cognitive function appears to be more likely improved by the neural activation than adversely affected by the cerebral perfusion. This thesis can be supported by the results obtained by Smale et al., who examined the influence of compression garments, which did not affect blood flow in the middle cerebral artery or exercise time trial performance, but improved psychomotor performance during exercise [39].

Even in energy-balanced conditions, resting energy expenditure is elevated 22 h after shorter sessions of both endurance and intermittent exercise [40]. The effects of physical activity are quantified as excess post-exercise oxygen consumption (EPOC), with short and prolonged components distinguished. The prolonged EPOC component lasts as long as 24 h depending on the intensity and duration of the previous exercise. It is linked to increased ventilation, cardiac output, body temperature and catecholamine concentration, resulting from metabolic changes within skeletal muscles, such as enhanced protein, glycogen and triacylglycerol resynthesis, respiratory uncoupling in mitochondria and upregulation of hypoxia-inducible factor 1α [41]. All the aforementioned factors increase the activity of the autonomic nervous system and produce afferent impulses that reach the brain and thus modify arousal in a multidirectional manner. The balance of those impulses is hard to predict at the present stage of knowledge and could be the possible explanation behind the diversity of psychomotor performance results observed in various studies. The key factor is a cumulative relative workload; if the overall intensity and duration of exercise fits within the sweet spot of individual tolerance, psychomotor performance would be maintained or improved, whereas if the load is too hard, psychomotor performance breaks down rapidly. In the SDE trial of the present study, the additional stimulation persisting after 30 h of prolonged exercise balanced the scale towards the more favorable arousal and at rest annihilated the usually observed detrimental effect of sleep deprivation. Therefore, the better psychomotor performance at rest in the SDE trial probably resulted from the enhanced neural activity of the brain due to 30 h of exercise and its metabolic consequences. 

During graded exercise, the reaction times were similar in both groups, but in the SDE trial, the fastest ones were observed at significantly lower workloads and the total number of missed reactions tended to be higher. In both groups, the U-shaped pattern of MCRT was confirmed, which is in agreement with most of the previous data [15,16,17,18]. The lack of differences in MCRT and the shortest MCRT between groups is in contrast to what would be expected based on the results of studies on SD alone, which almost unanimously report the detrimental effect of the lack of sleep on psychomotor performance and other aspects of the cognitive functions as well [2,3,4,5,19]. This could be explained by what has been already found at rest, namely the higher arousal induced by prolonged exercise had balanced its decrease produced by the lack of sleep. It works to some extent, especially during shorter tasks similar to those applied in the present study, namely the 107 s MCRT. In longer, more boring tasks involving attention, such balance is less likely to be achieved. Nevertheless, some detrimental effects of sleep deprivation were also found in the present study. First, the shift in the workload corresponding to the fastest MCRT tended towards lower values, which means that the optimal psychomotor performance of the subjects in the SDE trial was broken down significantly earlier. The second is the observed tendency for the total number of missed reactions to increase. Despite the good tolerance of the SDE trial by the participants, after 30 h of activity, fatigue definitely came into play. Therefore, due to cumulative central and peripheral fatigue, the corresponding workloads during the graded test were perceived as ‘harder’. The breakdown of reaction time observed at lower workloads in the SDE trial probably results from the altered lactate homeostasis above the anaerobic threshold leading to metabolic acidosis [16]. The rapid component of central fatigue originates from the afferent impulses from muscle chemoreceptors, which are stimulated by chemical changes in the extracellular milieu of skeletal muscles, especially by an exertional increase in the concentration of hydrogen ions.

This finding might have a practical implication; if exercise is used to stimulate arousal and thus enhance psychomotor performance, then in SDE subjects, it should be used more cautiously and lower workloads would probably enable a similar effect to be achieved, as the higher ones in well-slept individuals. Although stimulation by exercise can be recommended to enhance psychomotor performance in any group of affected people, special caution should be paid not to overload the SDE subjects.

Knowledge and awareness of the pattern of changes in psychomotor performance, resulting from both sleep deprivation and exercise, can be useful in planning the strategy of task implementation or enable its modification during operation. Moderate exercise of a short duration can be used or recommended to shortly minimize the detrimental effect of sleep deprivation.

The limitation of the study is the lack of the sleep deprivation trial alone performed on the same subjects; this was accepted due to logistical and human resource reasons. The other weakness is the use of just one psychological test, but only this was possible to perform during the graded exercise. However, more expanded cognition testing, including motor learning, would be possible at rest and should be considered in future studies.

## 5. Conclusions

Sleep deprivation combined with moderate, prolonged exercise induces slightly different outcomes compared to sleep deprivation alone. Trained, accustomed men are able to fulfil short tasks involving psychomotor abilities independently from some extent of central fatigue. The key factor determining the real human performance is likely the subtle balance between the stimulation by exercise and the detrimental effects of the lack of sleep. Moderate exercise can be used or recommended to shortly enhance psychomotor performance, including sleep-deprived subjects, in whom special caution is required in order not to overload them.

## Figures and Tables

**Figure 1 brainsci-13-00570-f001:**
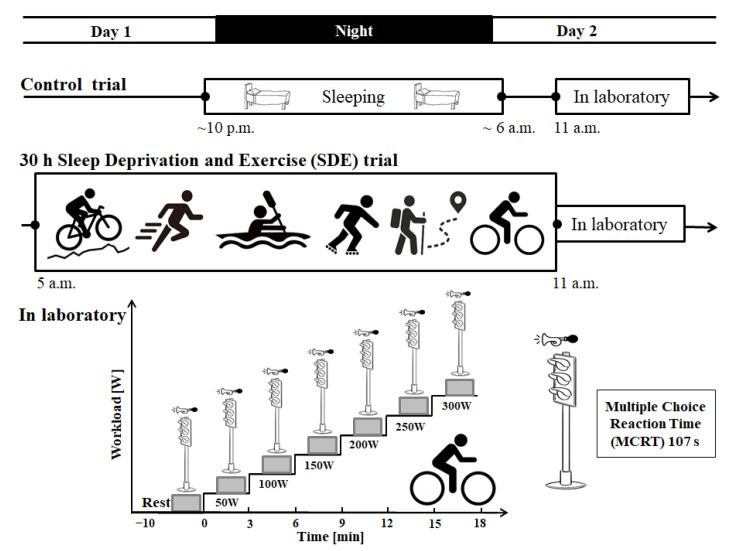
The block diagram of the experiment.

**Figure 2 brainsci-13-00570-f002:**
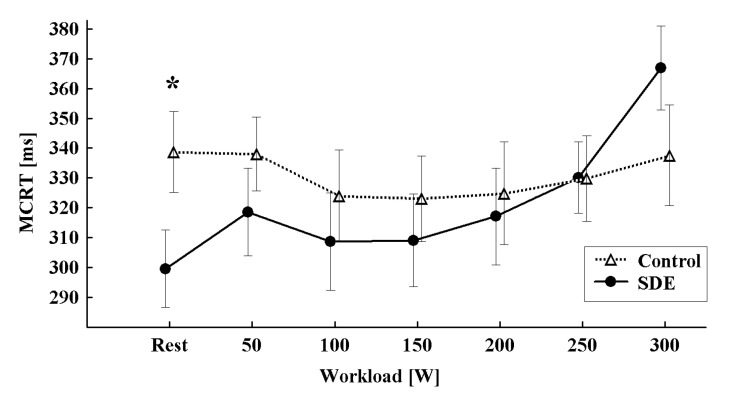
The multiple-choice reaction time (MCRT) during the graded exercise test to exhaustion in Control and after 30 h of sleep deprivation and exercise (SDE). Asterisk denotes a significant difference between Control and SDE trials; * *p* < 0.05.

**Figure 3 brainsci-13-00570-f003:**
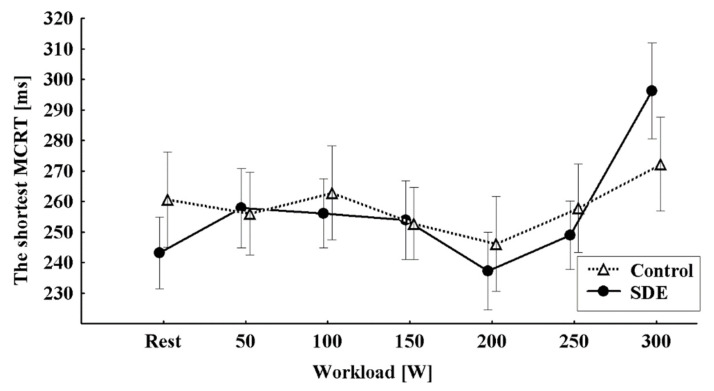
The shortest multiple-choice reaction times (MCRT) during the graded exercise test to exhaustion in Control and after 30 h of sleep deprivation and exercise (SDE).

## Data Availability

The data presented in this study are available on request from the corresponding author.

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
