# Peer review of "Psychomotor Performance after 30 h of Sleep Deprivation Combined with Exercise"

_brainsci, 2023, doi:10.3390/brainsci13040570_

Round 1

Reviewer 1 Report

It’s an interesting paper. I would love to see this paper get published after addressing the following issues: 

1.In the 2.4 Statistical Analysis section, can author elaborate more about what they used paired t-test and Wilcoxon test for?  Did the authors use Wilcoxon test because of the results from the Shapiro-Wilk test? If it does, what does Shapiro-wilk test results look like ? Do all variables meet the normality assumption ? 

2. in the 3 Results part, it will be really nice if the authors can also provide a plot for the missed reaction at different workload. Or, if on average the number of missed reaction is very low, the authors can report the minimum accuracy .

3. proof reading is recommended since some of the sentences are a little bit too long and hard to read. For example,  the ‘Eleven endurance-trained men with adventure racing experience performed …. (simulated adventure race). ‘ in the abstract.

Reviewer 2 Report

The submitted work by the authors examined the effects on psychomotor ability the sleep deprivation only (SD) and sleep deprivation coupled with a moderate exercise (SDE). Eleven subjects were recruited in this study. The authors showed that psychomotor ability is superior at rest following SDE, but not SD. This difference gradually deminished after each graded exercise when the workload increases. 

1) General comments:

Abstract section requires English editing. I have concerns about the Results section as it deserves more treatment and expansion. Discussion section is good.

2) Methods section: quite confusing. Please provide a block diagram of the experiment, like what usual behavioural studies do. The timeline and sequence of events can be confusing for readers! Did the authors perform any cognitive tests, e.g. working memory?

3) Please give more analysis to the Results section. The authors only provided the average MCRT. Is there any difference between positive and negative test results? If the # misses  is higher in SDE group, doesn't this mean their overall psychomotor ability is worse than SD group? Any insightful difference in heart rate between SD vs SDE? 

4) Was there any comments from the participants with regards to the trial (adverse effect, tolerance)? The authors recruited adults who are used to outdoor activities/adventure. Why didn't the authors recruit naive adults as control? Perhaps the authors can hypothetically discuss how this study will be with those adults. This can be important as working adults often have serious insomnia issue and work night-shifts. 

5) I think the SDE paradigm is rather interesting to add on to knowledge translation. Can the authors discuss the strength and limitation of this study? Future study: In addition to just psychomotor ability, can the authors study motor learning in SDE condition? 

Reviewer 3 Report

The manuscript of  Tomasz Mikulski et al. entitled „Psychomotor performance after 30 hours of sleep deprivation combined with exercise” provides interesting results about change in reaction time after 30 hours of complete sleep deprivation combined with prolonged exercise. The methods are well-described. I have some minor comments to improve the quality of the manuscript.

1. It should be mentioned, in which month(s) the examination was performed, including weather conditions. Was the weather the same in all cases?

2. VO2 max abbreviarion should be explained.

3. It should be indicated, what was the protocol when the heart rate monitor indicated a too high value.

4. Size of font should be checked throughout the manuscript, it is changing in various places.

Round 2

Reviewer 1 Report

all previous comments have been addressed 

Reviewer 2 Report

The authors have made major improvements to the manuscript. The limitation is stated. If interested, future studies can look into the impacts of SDE on motor learning (the authors can read more about the use of sequence learning tasks widely available in the literature).